# STRUCTURED PREDICTION USING cGANS WITH FUSION DISCRIMINATOR

**Faisal Mahmood, Wenhao Xu, Nicholas J. Durr**
Department of Biomedical Engineering
Johns Hopkins University
Baltimore, MD 21218 USA
`{faisalm,wenhao1,ndurr}@jhu.edu`

**Jeremiah W. Johnson**
Department of Applied Engineering & Sciences
University of New Hampshire
Manchester, NH 03101, USA
`jeremiah.johnson@unh.edu`

**Alan Yuille**
Department of Computer Science
Johns Hopkins University
Baltimore, MD 21218, USA
`ayuille1@jhu.edu`

## ABSTRACT

We propose the fusion discriminator, a single unified framework for incorporating conditional information into a generative adversarial network (GAN) for a variety of distinct structured prediction tasks, including image synthesis, semantic segmentation, and depth estimation. Much like commonly used convolutional neural network - conditional Markov random field (CNN-CRF) models, the proposed method is able to enforce higher-order consistency in the model, but without being limited to a very specific class of potentials. The method is conceptually simple and flexible, and our experimental results demonstrate improvement on several diverse structured prediction tasks.

## 1 INTRODUCTION

Convolutional neural networks (CNNs) have demonstrated groundbreaking results on a variety of different learning tasks. However, on tasks where high dimensional structure in the data needs to be preserved, per-pixel regression losses typically result in unstructured outputs since they do not take into consideration non-local dependencies in the data. Structured prediction frameworks such as graphical models and joint CNN-graphical model-based architectures *e.g.* CNN-CRFs have been used for imposing spatial contiguity using non-local information (Lin et al., 2016; Chen et al., 2018a; Schwing & Urtasun, 2015; Mahmood & Durr, 2018). The motivation to use CNN-CRF models stems from their ability to capture some structured information from second order statistics using the pairwise part. However, statistical interactions beyond the second-order are tedious to incorporate and render the models complicated (Arnab et al., 2016; Kohli et al., 2009).

Generative models provide another way to represent the structure and spacial contiguity in large high-dimensional datasets with complex dependencies. Implicit generative models specify a stochastic procedure to produce outputs from a probability distribution. Such models are appealing because they do not demand parametrization of the probability distribution they are trying to model. Recently, there has been great interest in CNN-based implicit generative models using autoregressive (Chen et al., 2018b) and adversarial training frameworks (Luc et al., 2016).

Generative adversarial networks (GANs) (Goodfellow et al., 2014) can be seen as a two player *minimax* game where the first player, the generator, is tasked with transforming a random input to a specific distribution such that the second player, the discriminator, can not distinguish between the true and synthesized distributions. The most distinctive feature of adversarial networks is the discriminator that assesses the discrepancy between the current and target distributions. The discriminator acts as a progressively precise critic of an increasingly accurate generator. Despite their structured prediction capabilities, such a training paradigm is often unstable. However, recent work on spectral normalization (SN) and gradient penalty has significantly increased training stability

(Miyato et al., 2018; Gulrajani et al., 2017). Conditional GANs (cGANs) (Mirza & Osindero, 2014) incorporate conditional image information in the discriminator and have been widely used for class conditioned image generation (Miyato et al., 2018; Miyato & Koyama, 2018). To that effect, unlike in standard GANs, a discriminator for cGANs discriminates between the generated distribution and the target distribution on pairs of samples $y$ and conditional information $x$.

For class conditioning, several unique strategies have been presented to incorporate class information in the discriminator (Reed et al., 2016; Miyato & Koyama, 2018; Odena et al., 2016).

However, a cGAN can also be conditioned by structured data such as an image. Such conditioning is much more useful for structured prediction problems. Since the discriminator in an image conditioned-GAN has access to large portions of the image the adversarial loss can be interpreted as a learned loss that incorporates higher order statistics, essentially eliminating the need to manually design higher order loss functions. This variation of cGANs has extensively been used for image-to-image translation tasks (Isola et al., 2017; Zhu et al., 2017). However, the best way of incorporating conditional image information into a GAN is not clear, and methods of feeding generated and conditional images to the discriminator tend to use a naive concatenation approach. In this work we address this gap by proposing a discriminator architecture specifically designed for image conditioning. Such a discriminator contributes to the promise of generalization that GANs bring to structured prediction problems by providing a singular and simplistic setup for capturing higher order non-local structural information from higher dimensional data without complicated modeling of energy functions.

Figure 1: Discriminator models for image conditioning. We propose fusing the features of the input and the ground truth or generated image rather than concatenating.

**Contributions.** We propose an approach to incorporating conditional information into a cGAN using a fusion discriminator architecture (Fig. 1b). In particular, we make the following key contributions:

1. We propose a novel discriminator architecture designed to incorporating conditional information for structured prediction tasks. The method is designed to incorporate conditional information in feature space in a way that allows the discriminator to enforce higher-order consistency in the model, and is conceptually simpler than alternative structured prediction methods such as CNN-CRFs where higher-order potentials have to be manually incorporated in the loss function.

2. We demonstrate the effectiveness of this method on a variety of distinct structured prediction tasks including semantic segmentation, depth estimation, and generating real images from semantic masks. Our empirical study demonstrates that the fusion discriminator is effective in preserving high-order statistics and structural information in the data and is flexible enough to be used successfully for many structured prediction tasks.

## 2 RELATED WORK

### 2.1 CNN-CRF MODELS

Models for structured prediction have been extensively studied in computer vision. In the past these models often entailed the construction of hand-engineered features. In 2015, Long et al. (2015) demonstrated that a fully convolutional approach to semantic segmentation could yield state-of-the-art results at that time with no need for hand-engineering features. Chen et al. (2014) showed

that post-processing the results of a CNN with a conditional Markov random field led to significant improvements. Subsequent work by many authors have refined this approach by incorporating the CRF as a layer within a deep network and thereby enabling the parameters of both models to be learnt simultaneously (Knöbelreiter et al., 2017). Many researchers have used this approach for other structured prediction problems, including image-to-image translation and depth estimation (Liu et al., 2015; Mahmood & Durr, 2018; Mahmood et al., 2018).

In most cases CNN-CRF models only incorporate unary and pairwise potentials. Arnab et al. (2016) investigated incorporating higher-order potentials into CNN-based models for semantic segmentation, and found that while it is possible to learn the parameters of these potentials, they can be tedious to incorporate and render the model quite complex. Thus there is a need to develop methods that can incorporate higher-order statistical information without requiring manual modeling of higher order potentials.

## 2.2 GENERATIVE ADVERSARIAL NETWORKS

**Adversarial Training.** Generative adversarial networks were introduced in Goodfellow et al. (2014). A GAN consists of a pair of models $(G, D)$, where $G$ attempts to model the distribution of the source domain and $D$ attempts to evaluate the divergence between the generative distribution $q$ and the true distribution $p$. GANs are trained by training the discriminator and the generator in turn, iteratively refining both the quality of the generated data and the discriminator's ability to distinguish between $p$ and $q$. The result is that $D$ and $G$ compete to reach a Nash equilibrium that can be expressed by the training procedure. While GAN training is often unstable and prone to issues such as mode collapse, recent developments such as spectral normalization and gradient penalty have increased GAN training stability (Miyato et al., 2018; Gulrajani et al., 2017). Furthermore, GANs have the advantage of being able to access the joint configuration of many variables, thus enabling a GAN to enforce higher-order consistency that is difficult to enforce via other methods (Luc et al., 2016; Isola et al., 2017).

**Conditional GANs.** A conditional GAN (cGAN) is a GAN designed to incorporate conditional information (Mirza & Osindero, 2014). cGANs have shown promise for several tasks such as class conditional image synthesis and image-to-image translation (Mirza & Osindero, 2014; Isola et al., 2017). There are several advantages to using the cGAN model for structured prediction, including the simplicity of the framework. Image conditioned cGANs can be seen as a structured prediction problem tasked with learning a new representation of an input image while making use of non-local dependencies. However, the method by which the conditional information should be incorporated into the model is often unmotivated. Usually, the conditional data is concatenated to some layers in the discriminator (often the input layers). A notable exception to this methodology is the projection cGAN, where the data is assumed to follow certain simple distributions, allowing a hard mathematical rule for incorporating conditional data can be derived from the underlying probabilistic graphical model (Miyato & Koyama, 2018). As mentioned in Miyato & Koyama (2018), the method is less likely to produce good results if the data does not follow one of the prescribed distributions. For structured prediction tasks involving conditioning with image data, this is often not the case. In the following section we introduce the fusion discriminator and explain the motivation behind it.

## 3 PROPOSED METHOD: CGANS WITH FUSION DISCRIMINATOR

As mentioned, the most significant part of cGANs for structured prediction is the discriminator. The discriminator has continuous access to pairs of the generated data or real data $y$ and the conditional information (*i.e.* the image) $x$. The cGAN discriminator can then be defined as $\mathcal{D}_{cGAN}(x, y, \theta) := \mathcal{A}(f(x, y, \theta))$, where $\mathcal{A}$ is the activation function, and $f$ is a function of $x$ and $y$ and $\theta$ represents the parameters of $f$. Let $p$ and $q$ designate the true and the generated distributions. The adversarial loss for the discriminator can then be defined as

$$\mathcal{L}(\mathcal{D}) = -\mathbb{E}_{q(y)}[\mathbb{E}_{q(x|y)} \log(\mathcal{D}(x, y, \theta)] - \mathbb{E}_{p(y)}[\mathbb{E}_{p(x|y)}[\log(1 - \mathcal{D}(x, G(x), \theta)]. \quad (1)$$

Here, $\mathcal{A}$ is the sigmoid function, $\mathcal{D}$ is the conditional discriminator, and $G$ is the generator. By design, this frameworks allows the discriminator to significantly effect the generator (Goodfellow

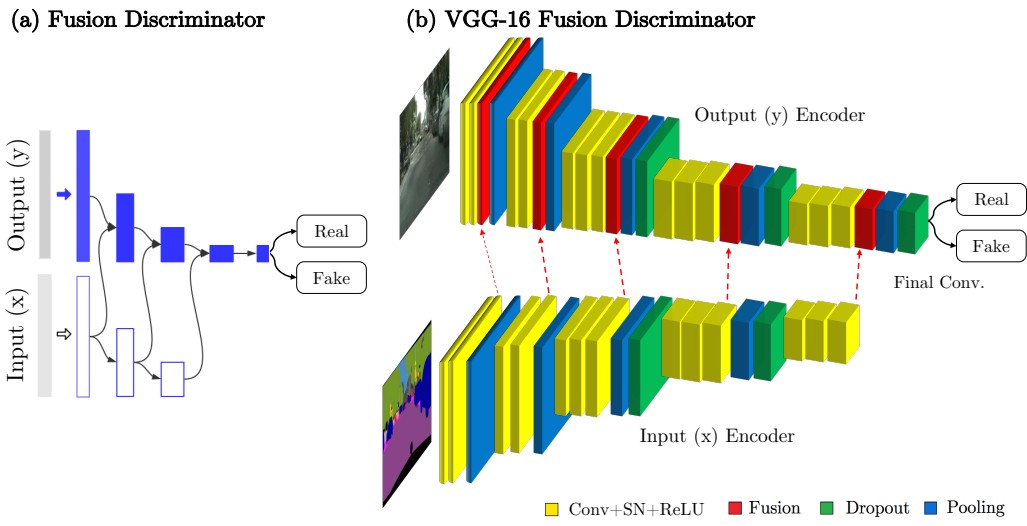

Figure 2: Fusion discriminator architecture.

et al., 2014). The most common approach currently in use to incorporate conditional image information into a GAN is to concatenate the conditional image information to the input of the discriminator at some layer, often the first (Isola et al., 2017). Other approaches for conditional information fusion are limited to class conditional fusion where conditional information is often a one-hot vector rather than higher dimensional structured data. Since the discriminator classifies pairs of input and output images, concatenating high-dimensional data may not exploit inherent dependencies in the structure of the data. Fusing the input and output information in an intuitive way such as to preserve the dependencies is instrumental in designing an adversarial framework with high structural capacity.

We propose the use of a fusion discriminator architecture with two branches. The branches of this discriminator are convolutional neural networks with identical architectures, say $\psi(x)$ and $\phi(y)$, that learn representations from both the conditional data ($\psi(x)$) and the generated or real data ($\phi(y)$) respectively. The learned representations are then fused at various stages (Fig. 2). This architecture is similar to the encoder portion of the FuseNet architecture, which has previously been used to incorporate depth information from RGB-D images for semantic segmentation (Hazirbas et al., 2017). In Figure 2, we illustrate a four layer and a VGG16-style fusion discriminator, in which both branches are similar in depth and structure to the VGG16 model (Simonyan & Zisserman, 2014). The key ingredient of the fusion discriminator architecture is the fusion block, which combines the learned representations of $x$ and $y$. The fusion layer (red, Fig. 2) is implemented as element-wise summation and is always inserted after a convolution $\rightarrow$ spectral normalization $\rightarrow$ ReLU instance. The fusion layer modifies the signal passed through the $\psi$ branch by adding in learned representations of $x$ from the $\phi$ branch. This preserves representation from both $x$ and $y$. For structured prediction tasks, $x$ and $y$ will often have learned representations that complement each other; for instance, in tasks like depth estimation, semantic segmentation, and image synthesis, $x$ and $y$ all have highly complimentary features.

### 3.1 MOTIVATION

**Theoretical Motivation.** When data is passed through two networks with identical architectures and the activations at corresponding layers are added, the effect is to pass through the combined network (the upper branch in Fig. 2) a stronger signal than would be passed forward by applying an activation to concatenated data.

To see this in the case of the ReLU activation function, denote the $k^{th}$ feature map in the $l^{th}$ layer by $\mathbf{h}_k^{(l)}$ and let the weights and biases for this feature and layer be denoted $W_K^{(l)} = [U_k^{(l)} \ V_K^{(l)}]^T$ and $b_k^{(l)} = [c_k^{(l)} \ d_k^{(l)}]^T$ respectively. Let $\mathbf{h} = \begin{bmatrix} \mathbf{x}^T \ \mathbf{y}^T \end{bmatrix}^T$, where $\mathbf{x}$ and $\mathbf{y}$ represent the learned features from the conditional and real or generated data respectively. Then

$$\mathbf{h}_k^{(l+1)} = \max(\mathbf{0}, W_k^{(l)}\mathbf{h} + b_k^{(l)}) \tag{2}$$

$$= \max(\mathbf{0}, (U_k^{(l)}\mathbf{x}^{(l)} + V_k^{(l)}\mathbf{y}^{(l)} + (c_k^{(l)} + d_k^{(l)}))) \tag{3}$$

$$\leq \max(\mathbf{0}, (U_k^{(l)}\mathbf{x}^{(l)} + c_k^{(l)})) + \max(\mathbf{0}, (V_k^{(l)}\mathbf{y}^{(l)} + d_k^{(l)})). \tag{4}$$

Eq. 4 demonstrates that the fusion of the activations in $\psi(x)$ and $\phi(y)$ produces a stronger signal than the activation on concatenated inputs.[1] Strengthening some activations does not guarantee improved performance in general; however, in the context of structured prediction the fusing operation results in the strongest signals being passed through the discriminator specifically at those places where the model finds useful information simultaneously in both the conditional data and the real or generated data.

A similar mechanism can be found at at work in many other successful models that require higher order structural information to be preserved; to take one example, consider the neural algorithm of artistic style proposed by Gatys et al. (2015). This algorithm successfully transfers highly structured data from an existing image $\mathbf{x}$ onto a randomly initialized image $\mathbf{y}$ by minimizing the content loss function

$$\mathcal{L}_{content}(\mathbf{x}, \mathbf{y}, l) = \frac{1}{2}\sum_{i,j}\left(F_{ij}^l - P_{ij}^l\right)^2 , \tag{5}$$

where $F_{ij}^l$ and $P_{ij}^l$ denote the activations at locations $i, j$ in layer $l$ of $\mathbf{x}$ and $\mathbf{y}$ respectively. The loss function mechanism used here differs from the fusing mechanism used in the fusion discriminator, but the underlying principle of capturing high-level structural information from a pair of images by combining signals from common layers in parallel networks is the same. The neural algorithm of artistic style succeeds in content transfer by insuring that the activations containing information of structural importance is similar in both the generated image and the content image. In the case of image-conditioned cGAN training, it can be assumed that the activations of the real or generated data and the conditional data will be similar, and by fusing these activations and passing forward a strengthened signal the network is better able to attend to those locations containing important structural information in both the real or generated data and the conditional data; c.f. Fig. 3.

**Empirical Motivation.** We use gradient-weighted Class Activation Mapping (Grad-CAM) (Selvaraju et al., 2017) which uses the class-specific gradient information going into the final convolutional layer of a trained CNN to produce a coarse localization map of the important regions in the image. We visualized the outputs of a fusion and concatenated discriminator for several different tasks to observe the structure and strength of the signal being passed forward. We observed that the fusion discriminator architecture always had a visually strong signal at important features for the given task. Representative images from classifying $\mathbf{x}$ and $\mathbf{y}$ pairs as 'real' for two different structured prediction tasks are shown in Fig. 3. This provides visual evidence that a fusion discriminator preserves more structural information from the input and output image pairs and classifies overlapping patches based on that information. Indeed, this is not evidence that a stronger signal will lead to a more accurate classification, but it is a heuristic justification that more representative features from $\mathbf{x}$ and $\mathbf{y}$ will be used to make the determination.

## 4    EXPERIMENTS

In order to evaluate the effectiveness of the proposed fusion discriminator we conducted three sets of experiments on structured prediction problems: 1) generating real images from semantic masks (Cityscapes); 2) semantic segmentation (Cityscapes); 3) depth estimation (NYU v2). For all three tasks we used a U-Net based generator. We applied spectral normalization to all weights of the generator and discriminator to regularize the Lipschitz constant. The Adam optimizer was used for all experiments with hyper-parameters $\alpha = 0.0002$, $\beta_1 = 0$, $\beta_2 = 0.9$.

---

[1]Equations 2–4 apply only for the ReLU, but similar statements can be easily proven for many commonly used activation functions; see Section 6 for additional discussion.

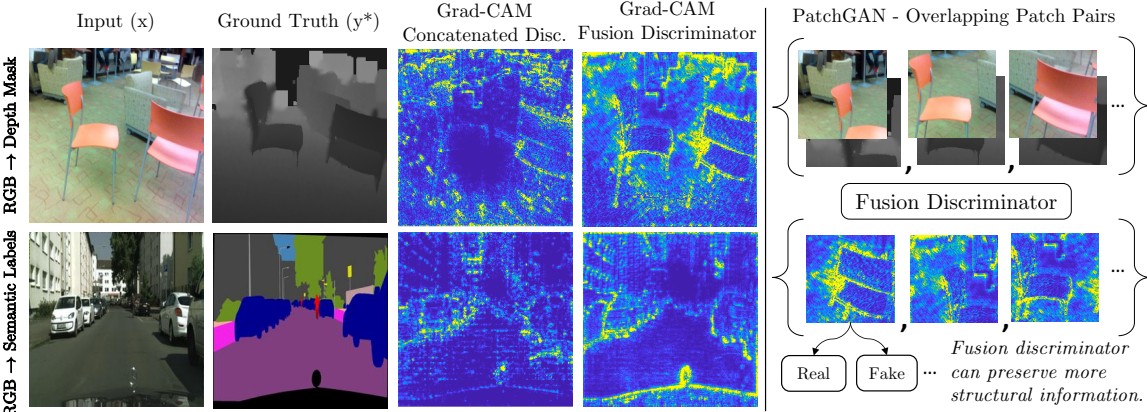

Figure 3: Visualizing Discriminator features using gradient-weighted Class Activation Maps (Grad-CAM) to produce a coarse localization map of the important regions in the image. The fusion discriminator passes a stronger and more structured signal on important features in comparison to a concatenated discriminator.

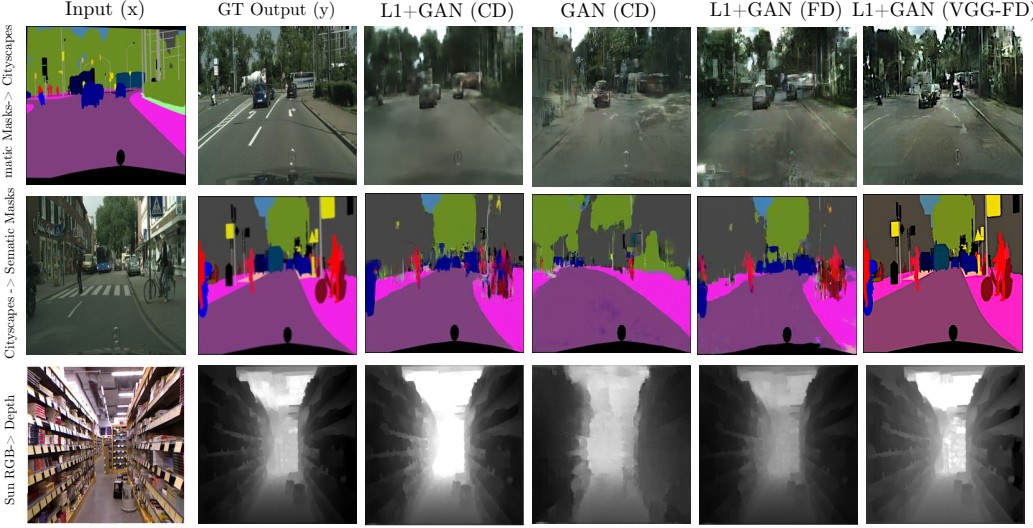

Figure 4: Comparative analysis of concatenation and fusion discriminators on three different structured prediction tasks, a) Semantic masks to real image transformation b) Semantic segmentation c) Depth Estimation. The fusion discriminator preserves more structural details.

## 4.1 IMAGE-TO-IMAGE TRANSLATION

In order to demonstrate the structure preserving abilities of our discriminator we use the proposed setup in the image-to-image translation setting. We focus on the application of generating realistic images from semantic labels. This application has recently been studied for generating realistic synthetic data for self driving cars (Wang et al., 2018; Chen & Koltun, 2017). Unlike recent approaches where the objective is to generate increasingly realistic high definition (HD) images, the purpose of this experiment is to explore if a generic fusion discriminator can outperform a concatenated discriminator when using a simple generator. We used 2,975 training images from the Cityscapes dataset (Cordts et al., 2016) and re-scaled them to $256 \times 256$ for computational efficiency. The provided Cityscapes test set with 500 images was used for testing. Our ablation study focused on changing the discriminator between a standard 4-layer concatenation discriminator used in the sem-

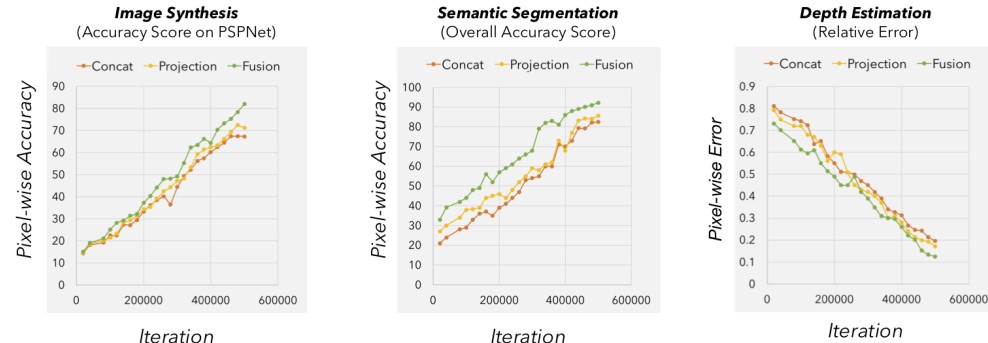

Figure 5: A comparative analysis of concatenation, projection and fusion discriminators on three different structured prediction tasks, *i.e.*, image synthesis, semantic segmentation, and depth estimation.

Table 1: PSPNet-based semantic segmentation IoU and accuracy scores using generated images from different discriminators. Our results outperform concatenation-based methods by a large margin and is close to the accuracy and IoU on actual images (GT/Oracle).

| Discriminator | Mean IoU | Pixel Accuracy |
|---|---|---|
| 4-Layer Concat. (Isola et al. (2017)) | 0.3617 | 74.34% |
| 4-Layer Concat. + SN | 0.4022 | 76.49% |
| 4-Layer Fusion + SN | 0.4569 | 79.23% |
| VGG16 Concat. + SN | 0.4125 | 77.62% |
| Projection + SN (Miyato & Koyama (2018)) | 0.4696 | 79.11% |
| **VGG16 Fusion + SN** | **0.5483** | **83.07%** |
| GT / Oracle | 0.5937 | 85.13% |

inal image-to-image translation work (Isola et al., 2017), a combination of this 4-layer discriminator with spectral normalization (SN) (Miyato et al., 2018), a VGG-16 concatenation discriminator and the proposed 4-layer and VGG-16 fusion discriminators.

### 4.1.1 EVALUATION

Since standard GAN evaluation metrics such as inception score and FID can not directly be applied to image-to-image translation tasks we use an evaluation technique previously used for such image synthesis Isola et al. (2017); Wang et al. (2017). To quantitatively evaluate the effectiveness of our proposed discriminator architecture we perform semantic segmentation on synthesized images and compare the similarity between the predicted segments and the input. The intuition behind this kind of experimentation is that if the generated images corresponds to the input label map an existing semantic segmentation model such as a PSPNet (Zhao et al., 2017) should be able to predict the input segmentation mask. Similar experimentation has been suggested in Isola et al. (2017) and Wang et al. (2017). Table 1 reports segmentation both pixel-wise accuracy and overall intersection-over-union (IoU). The proposed fusion discriminator outperforms the concatenated discriminator by a large margin. Our result is closer to the theoretical upper bound achieved by real images. This confirms that the fusion discriminator contributes to structure preservation in the output image. The fusion discriminator could be used with high definition images, however, such analysis is beyond the scope of the current study. Representative images for this task are shown in Fig. 4. The projection discriminator was modified image conditioning according to the explanation given in Miyato & Koyama (2018) for the super-resolution task. Fig. 5 shows a comparative analysis of the concatenation, projection and fusion discriminators in an ablation study upto $550k$ iterations.

Table 2: GAN-based semantic segmentation using different discriminators. Tested with cityscapes dataset rescaled to $256 \times 256$ images.

| Discriminator | Mean IoU | Pixel Accuracy |
|---|---|---|
| 4-Layer Concat. (Isola et al. (2017)) | 0.2925 | 81.41% |
| 4-Layer Concat. + SN | 0.3162 | 83.49% |
| 4-Layer Fusion + SN | 0.4471 | 85.23% |
| VGG16 Concat. + SN | 0.4066 | 84.62% |
| Projection + SN (Miyato & Koyama (2018)) | 0.4687 | 85.97% |
| **VGG16 Fusion + SN** | **0.6642** | **92.17%** |
| CNN-CRF Postprocess | 0.5425 | 87.41% |
| CNN-CRF Joint Training | 0.6042 | 90.25% |

## 4.2 SEMANTIC SEGMENTATION

Semantic segmentation is vital for visual scene understanding and is often formulated as a dense labeling problem where the objective is to predict the category label for each individual pixel. Semantic segmentation is a classical structured prediction problem and CNNs with pixel-wise loss often fail to make accurate predictions (Luc et al., 2016). Much better results have been achieved by incorporating higher order statistics in the image using CRFs as a post-processing step or jointly training them with CNNs (Chen et al., 2018a). It has been shown that incorporating higher order potentials continues to improve semantic segmentation improvement, making this an ideal task for evaluating the structured prediction capabilities of GANs and their enhancement using our proposed discriminator.

Here, we empirically validate that the adversarial framework with the fusion discriminator can preserve more spacial context in comparison to CNN-CRF setups. We demonstrate that our proposed fusion discriminator is equipped with the ability to preserve higher order details. For comparative analysis we compare with relatively shallow and deep architectures for both concatenation and fusion discriminators. We also conduct an ablation study to analyze the effect of spectral normalization. The generator for all semantic segmentation experiments was a U-Net. For the experiment without spectral normalization, we trained each model for 950k iterations, which was sufficient for the training of the concatenated discriminator to stabilize. For all other experiments, we trained for 800k iterations. The discriminator was trained twice as much as the generator.

## 4.3 DEPTH ESTIMATION

Depth estimation is another structured prediction task that has been extensively studied because of its wide spread applications in computer vision. As with semantic segmentation, both per-pixel losses and non-local losses such as CNN-CRFs have been widely used for depth estimation. State-of-the art with depth estimation has been achieved using a hierarchical chain of non-local losses. We argue that it is possible to incorporate higher order information using a simple adversarial loss with a fusion discriminator.

In order to validate our claims we conducted a series of experiments with different discriminators, similar to the series of experiments conducted for semantic segmentation. We used the *Eigen* test-train split for the NYU v2 Nathan Silberman & Fergus (2012) dataset containing 1449 images for training and 464 images for testing. We observed that as with image synthesis and semantic segmentation the fusion discriminator outperforms concatenation-based methods and pairwise CNN-CRF methods every time.

## 5 CONCLUSIONS

Structured prediction problems can be posed as image conditioned GAN problems. The discriminator plays a crucial role in incorporating non-local information in adversarial training setups for structured prediction problems. Image conditioned GANs usually feed concatenated input and output pairs to the discriminator. In this research, we proposed a model for the discriminator of cGANs that involves fusing features from both the input and the output image in feature space. This method

Table 3: Depth Estimation results on NYU v2 dataset using various discriminators.

| Discriminator | relative error | rms | $\log_{10}$ |
|---|---|---|---|
| 4-Layer Concat. (Isola et al. (2017)) | 0.1963 | 0.784 | 0.087 |
| 4-Layer Concat. + SN | 0.1442 | 0.592 | 0.059 |
| 4-Layer Fusion + SN | 0.1315 | 0.583 | 0.057 |
| VGG16 Concat. + SN | 0.1374 | 0.547 | 0.054 |
| Projection + SN (Miyato & Koyama (2018)) | 0.1417 | 0.573 | 0.059 |
| **VGG16 Fusion + SN** | **0.1254** | **0.491** | **0.052** |
| CNN-CRF Postprocess | 0.311 | 1.025 | 0.129 |
| CNN-CRF Joint Training | 0.232 | 0.824 | 0.094 |

provides the discriminator a hierarchy of features at different scales from the conditional data, and thereby allows the discriminator to capture higher-order statistics from the data. We qualitatively demonstrate and empirically validate that this simple modification can significantly improve the general adversarial framework for structured prediction tasks. The results presented in this paper strongly suggest that the mechanism of feeding paired information into the discriminator in image conditioned GAN problems is of paramount importance.

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

# 6 SUPPLEMENTARY MATERIAL

## 6.1 CGAN OBJECTIVE

The objective function for a conditional GANs can be defined as,

$$\mathcal{L}_{cGAN}(G, D) = \mathbb{E}_{x,y}[\log(D(x, y)] + \mathbb{E}_{x,z}[\log(1 - D(x, G(x))]. \tag{6}$$

The generator $G$ tries to minimize the loss expressed by equation 6 while the discriminator $D$ tries to maximize it. In addition, we impose an $L1$ reconstruction loss:

$$\mathcal{L}_{L1}(G) = \mathbb{E}_{x,y}[||y - G(x)||_1], \tag{7}$$

leading to the objective,

$$G^* = \arg\min_G \max_D \mathcal{L}_{cGAN}(G, D) + \lambda \mathcal{L}_{L1}(G). \tag{8}$$

## 6.2 GENERATOR ARCHITECTURE

We adapt our network architectures from those explained in (Isola et al., 2017). Let CSRk denote a Convolution-Spectral Norm -ReLU layer with $k$ filters. Let CSRDk donate a similar layer with dropout with a rate of 0.5. All convolutions chosen are $4 \times 4$ spatial filters applied with a stride 2, and in decoders they are up-sampled by 2. All networks were trained from scratch and weights were initialized from a Gaussian distribution of mean 0 and standard deviation of 0.02. All images were cropped and rescaled to $256 \times 256$, were up sampled to $268 \times 286$ and then randomly cropped back to $256 \times 256$ to incorporate random jitter in the model.

**Encoder**: CSR64→CSR128→CSR256→CSR512→CSR512→CSR512→CSR512→CSR512
**Decoder**: CSRD512→CSRD1024→CSRD1024→CSR1024→CSR1024→CSR512→CSR256
→CSR128

The last layer in the decoder is followed by a convolution to map the number of output channels (3 in the case of image synthesis and semantic labels and 1 in the case of depth estimation). This is followed by a *Tanh* function. Leaky ReLUs were used throughout the encoder with a slope of 0.2, regular ReLUs were used in the decoder. Skip connections are placed between each layer $l$ in the encoder and layer $ln$ in the decoder assuming $l$ is the maximum number of layers. The skip connections concatenate activations from the $l^{th}$ layer to layer $(l - n)^{th}$ later.

## 6.3 ACTIVATIONS WITH NEGATIVE BRANCHES

Equations 2–4 of section 3.1 illustrate that when the ReLU activation is used in a fusion block, the fusing operation results in a positive signal at least as large as that obtained by concatenation. For activations with negative branches, the following similar claim holds.

**Lemma 1** *Denote the $k^{th}$ feature map in the $l^{th}$ layer by $\mathbf{h}_k^{(l)}$, and let the weights and biases for this feature and layer be denoted $W_K^{(l)} = [U_k^{(l)} \ V_K^{(l)}]^T$ and $b_k^{(l)} = [c_k^{(l)} \ d_k^{(l)}]^T$ respectively. Let $\mathbf{h} = [\mathbf{x}^T \ \mathbf{y}^T]^T$, where $\mathbf{x}$ and $\mathbf{y}$ represent the learned features from the conditional and real or generated data respectively. Let $\sigma$ represent an activation function. If $\text{sign}(\sigma(U_k^{(l)}\mathbf{x}^{(l)} + c_k^{(l)})) = \text{sign}(\sigma(V_k^{(l)}\mathbf{y}^{(l)} + d_k^{(l)}))$, then*

$$|\sigma(W_k^{(l)}\mathbf{h} + b_k^{(l)})| \leq |\sigma(U_k^{(l)}\mathbf{x}^{(l)} + c_k^{(l)})| + |\sigma(V_k^{(l)}\mathbf{y}^{(l)} + d_k^{(l)})|. \tag{9}$$

**Proof 2** *Trivial; c.f. equations 2–4.*

However, the general situation can be much more complex. Ideally, if $\text{sign}(\sigma(U_k^{(l)}\mathbf{x}^{(l)} + c_k^{(l)})) \neq \text{sign}(\sigma(V_k^{(l)}\mathbf{y}^{(l)} + d_k^{(l)}))$, then $|\sigma(W_k^{(l)}\mathbf{h} + b_k^{(l)})| \geq |\sigma(U_k^{(l)}\mathbf{x}^{(l)} + c_k^{(l)})| + |\sigma(V_k^{(l)}\mathbf{y}^{(l)} + d_k^{(l)})|$, but this claim cannot be made in general. A counterexample is given by the leaky ReLU function $\sigma(x) = \max(0, x) - \alpha \max(0, -x)$, where $\alpha \in \mathbb{R}^+$. In the case where $\sigma(U_k^{(l)}\mathbf{x}^{(l)} + c_k^{(l)})) \geq 0$ and

$\sigma(V_k^{(l)}\mathbf{y}^{(l)} + d_k^{(l)}) \leq 0$, fusing leads to the activation $U_k^{(l)}\mathbf{x}^{(l)} + c_k^{(l)} + \alpha(V_k^{(l)}\mathbf{y}^{(l)} + d_k^{(l)})$, while concatenation results in the activation $-\alpha(U_k^{(l)}\mathbf{x}^{(l)} + c_k^{(l)} + V_k^{(l)}\mathbf{y}^{(l)} + d_k^{(l)})$. The value of $\alpha$ plays a significant role in shaping the combined activation, and in some instances fusing can lead to a stronger signal than concatenation despite the disagreement in the incoming signals.

