# OpenReview forum: "Structured Prediction using cGANs with Fusion Discriminator"
_ICLR.cc/2019/Workshop/DeepGenStruct — DeepGenStruct 2019_

### Official Review · AnonReviewer1 · 2019-04-16
**discriminator fuses input/generator output at featuremap levels**

**Rating:** 2
**Confidence:** 2

**Review:**

In this paper, the problem of conditional image generation with conditional GANs is considered. The discriminator needs to take in both the input being conditioned on, as well as the generator output. Traditionally, the input and generator output are concatenated before passing into the discriminator network, while in this work, the input/generator output get passed through CNNs (with identical parameters) separately, and the features get merged at various levels. The authors argued that 1) theoretically the signal of fusing afterward is stronger than passing beforehand under certain conditions, and 2) empirically this discriminator architecture works better by a large margin on image synthesis, semantic segmentation and depth estimation.

Pros:
1. This paper is very clear and easy to follow.
2. The results show that the proposal gets better performance than concatenation by a large margin.

Cons:
1. As pointed out by this paper, a stronger signal does not necessarily mean better performance, hence I think the theoretical results do not make much sense here. Although the results seem promising, I am not sure if these results generalize to other tasks of varying dataset sizes and model architectures.
2. The novelty of this approach seems to be limited, and it's not clear how this approach can generalize to tasks other than image-to-image.

---

### Official Review · AnonReviewer2 · 2019-04-17
**fusion discriminator for conditional GAN**

**Rating:** 3
**Confidence:** 2

**Review:**

This paper presents a fusion discriminator used in conditional image generation based on GANs. Specifically, instead of taking the concatenated representations of the condition image (input) and the generated image (output), the discriminator encodes the two image by "fusion", i.e. intermediate representations from the condition image encoder and the generated image encoder are combined (e.g. summed).

The paper is well-written. The approach is reasonable and the results look good. The main concern is that the contribution is very focused: the idea of fusion is not new and applying it to cGAN seems to be a small contribution.

---

### Decision · Program_Chairs · 2019-04-19
**Acceptance Decision**

**Decision:**

Accept

**Comment:**

This paper presents a fusion discriminator which conditions upon extra information through fusion at multiple layers of the discriminator. The paper is well-executed and the experiments are convincing.